# Chemical and Physical Characterisation of Human Serum Albumin Nanocolloids: Kinetics, Strength and Specificity of Bonds with ^99m^Tc and ^68^Ga

**DOI:** 10.3390/nano11071776

**Published:** 2021-07-08

**Authors:** Manuela Marenco, Letizia Canziani, Gianluca De Matteis, Giorgio Cavenaghi, Carlo Aprile, Lorenzo Lodola

**Affiliations:** 1Fondazione IRCCS Policlinico San Matteo, Nuclear Medicine Unit, 27100 Pavia, Italy; g.dematteis@smatteo.pv.it (G.D.M.); g.cavenaghi@smatteo.pv.it (G.C.); l.lodola@smatteo.pv.it (L.L.); 2National Center for Oncological Hadrontherapy (CNAO), 27100 Pavia, Italy; carlo.aprile@icloud.com

**Keywords:** radiopharmaceutical, PET (positron emission tomography), SPECT (single-photon emission computed tomography), nanocolloids, human serum albumin, ^99m^Tc, ^68^Ga

## Abstract

Nanoparticles of Human Serum Albumin (NC) labelled with ^99m^Tc are widely used in Nuclear Medicine and represent the gold-standard for the intraoperative detection of the sentinel lymph node in many kinds of cancer, mainly breast cancer and melanoma. A significant amount of radionuclides can be incorporated into the HSA particle, due to the multiple binding sites, and HSA-based nanocolloid catabolism is a fast and easy process that results in innocuous degradation products. NCs labelled with different isotopes represent an interesting radiopharmaceutical for extending diagnostic accuracy and surgical outcome, but the knowledge of the chemical bond between NCs and isotopes has not been fully elucidated, including information on its strength and specificity. The aim of this study is to investigate and compare the physicochemical characteristics of the bond between NCs and ^99m^Tc and ^68^Ga isotopes. Commercial kits of HSA-based nanocolloid particles (NanoAlbumon^®^) were used. For this purpose, we have primarily studied the kinetic orders of NC radiolabelling. Langmuir isotherms and pH effect on radiolabelling were tested and the stability of the radiometal complex was verified through competition reactions carried out in presence of different ligands. The future goal of our research is the development of inexpensive and instant kits, easily labelled with a wide spectrum of diagnostic and therapeutic isotopes, thus facilitating the availability of versatile and multipurpose radiopharmaceuticals.

## 1. Introduction

Nuclear Medicine is based on the administration of peculiar compounds, the so-called radiopharmaceuticals, that are composed of a radionuclide (such as ^99m^Tc, ^18^F, ^68^Ga, ^223^Ra, ^64^Cu, etc.) chelated by a suitable ligand and conjugated to a molecule (such as a metabolic intermediate, a human protein fragment or colloid) acting as a vector able to transport the compound from the injection site to the region of interest. The vector investigated in this study is Nanocolloid Human Serum Albumin (NC), a colloidal suspension of HSA particles. Nanoparticles of HSA labelled with ^99m^Tc ([^99m^Tc]Tc-NC) are widely used as a radiopharmaceutical in Nuclear Medicine and represent the European gold-standard for the intraoperative detection of the Sentinel Lymph Node (SLN) in many kinds of cancer, mainly breast cancer and melanoma [1,2,3,4]. The associated favourable γ-emission of ^99m^Tc allows a preoperative check of colloid migration either with dynamic or static studies employing planar scintigraphy or hybrid instruments, such as SPECT-CT (single-photon emission computed tomography—computed tomography scan).

[^99m^Tc]Tc-NC is helpful in SLN-SPECT because of its ability to map nodes on the tumour drainage pathway, indicating the tumour state of the whole regional lymphatic basin and excluding lymphatic metastatic spread, in case of negative biopsy. NC can be used for the evaluation of pulmonary ventilation in patients with lung damage due to Covid-19 pneumonia [5,6].

Even though the intralymphatic kinetics of NC radiopharmaceuticals are not well understood, their ability to visualise the first lymph node of a chain is widely recognised, as in SLN detection for breast-conservating surgery for breast cancer.

Recent developments in precision medicine, such as evaluating regional perfusion in healthy versus cancerous tissues, points to the need for multi-modal contrast agents that can provide both better cross-sectional images via SPECT and positron emission tomography (PET). [7,8].

Furthermore, the increasingly common use of robotic surgery has highlighted the need for a specific procedure which can identify the site and contours of the target lesion [9].

For this reason, the research interest is now increasingly focused on the possibility of labelling commercial kits routinely used with other isotopes, to extend diagnostic accuracy, surgical outcome, follow-up and even therapeutic horizons.

Our previous study concerned the in vitro feasibility of obtaining an instant hybrid NC tracer, radiolabelled with both ^99m^Tc and ^68^Ga isotopes and tagged with the fluorescent probe Indocyanine Green (ICG) nanocolloids (NC) ([^99m^Tc]Tc-[^68^Ga]Ga-ICG-NC) [10]. The resulting tri-modal [^99m^Tc]Tc-[^68^Ga]Ga-ICG-NC combined three different imaging system (SPECT, PET and fluorescence) in a single tracer, thus responding to different surgical needs and improving the clinical application of the NC commercial kits.

However, before transferring its use to patients, it is mandatory to increase our knowledge of the chemical bond between NC and the isotopes. Currently, too little is known about the chemical conformation of the NC, even less about the strength and the specificity of the bond with Tc and almost nothing is known about the bond with Ga.

The aim of this study is to investigate and compare the chemical bond between NC and different isotopes. For this purpose, we have primarily studied the kinetic orders of NC labelling with ^99m^Tc and ^68^Ga, and then compared them with the kinetic orders labelled HSA.

Langmuir isotherms then made it possible to find the maximum concentration of complexable radiometal.

Finally, the bond strength and specificity were indirectly investigated by changing the starting conditions. The bond modification possibility was investigated with the simultaneous presence of the same quantities of the two isotopes. The pH effect on radiolabelling was tested and the stability of the radiometals complex was verified through competition reactions carried out in presence of a different ligand.

Therefore, HSA-based nanocolloids represent an interesting carrier, since a significant amount of radionuclides can be incorporated into the HSA particle, because of the several binding sites of albumin. Moreover, NC catabolism is a fast and easy process that results in innocuous degradation products [11].

## 2. Materials and Methods

### 2.1. Materials

All the pharmaceuticals used in this study have already been commercialised and authorised for clinical use. Nanocolloid (nano-sized human colloidal particles of a ≤80 nm diameter, containing 0.5 mg of human albumin) is a radiopharmaceutical commercially available as NanoAlbumon^®^ (Radiopharmacy Laboratory Ltd, Budaörs, Hungary). The NanoAlbumon^®^ kit is a sterile, non-pyrogenic, lyophilised mixture. Active ingredient: Human Serum Albumin nano sized colloid 0.5 mg. Excipients: stannous (II) chloride dihydrate 0.2 mg, sodium phosphate monobasic and sodium phosphate dibasic 1.0 mg, glucose 15.0 mg. In this article, the NanoAlbumon^®^ kit was generically referred as NC.

Human Serum Albumin is a radiopharmaceutical commercially available as Vasculocis^®^ containing 10 mg of human serum albumin, stannous chloride dihydrate, hydrochloric acid and sodium chloride, under nitrogen atmosphere. (CURIUM IBA Cis-Bio, Gif-sur-Yvette, Paris, France). In this article, Vasculocis^®^ was generically referred as HSA.

Dimercaptosuccinic acid (DMSA) is a radiopharmaceutical commercially available as Renocis^®^, containing 1 mg of dimercaptosuccinic acid, stannous chloride dihydrate, inositol, ascorbic acid. In this article, the Renocis^®^ kit was generically referred as DMSA.

All kits were labelled with sodium pertechnetate [^99m^TcO_4_^−^], obtained from ^99^Mo/^99m^Tc Tekcis® generator (CURIUM IBA Cis-Bio, Gif-sur-Yvette, Paris, France) and sodium chloride per injection (0.9%). To perform [^68^Ga]Ga labelling, ^68^Ge/^68^Ga generator (1.1 GBq TiO_2_-based GalliaPharm® Eckert-Ziegler Isotope Products, Berlin, Germany) was eluted with 8 mL of 0.1 N HCl (Eckert-Ziegler). 0.75 mL of 0.1 N NaOH / phosphate buffer (1 mL) was added to this 0.1N HCl solution of [^68^Ga] Ga-chloride. Final pH ranged between 4.0 and 4.5.

All chemicals were manufactured by TraceSELECT—UltraPURE from ABX Radensberg, Germany.

### 2.2. Methods

#### 2.2.1. Quality Controls and Labelling Yield

Quality controls (QC) were performed to verify the labelling yield of [^99m^Tc]Tc-NC and [^68^Ga]Ga-NC. The percentage of free ^99m^Tc was evaluated by thin-layer chromatography, using ITLC-SG (Varian, Folson, CA, USA) as a stationary phase (10 cm long and 2 cm wide) and CH_3_OH:H_2_O 85:15 as a mobile phase. A 10 μL spot of solution containing the sample was applied to a strip roughly 1.5 cm from the bottom edge. The strip was then placed in a separation chamber and the solvent was run for at least 10 cm. Labelled NC remains at the point of application, while free ^99m^Tc pertechnetate migrates with the solvent front. The percentage of free ^68^Ga was assessed in the same way, using 0.1 M tribasic-citrate solution, adjusted to pH 6 with HCl, as a mobile phase. NC labelled with ^68^Ga remains at the point of application, while free ^68^Ga migrates with the solvent front. TLC (thin layer chromatography) strips were cut at 4.5 cm in height, defining a down sample region with point of application, and an up sample with the solvent front. In these two regions, ^99m^Tc and ^68^Ga activity was determined with a 3″ × 3″ NaI(Tl) pinhole 16 × 40 mm gamma counter (Raytest, Straubenhardt, Germany).

#### 2.2.2. [^99m^Tc]Tc-HSA Labelling Kinetics

According to the manufacturers’ instructions, the HSA kit was reconstituted in 4 mL of saline solution with 2000 MBq of sodium pertechnetate [^99m^TcO_4_^−^] obtained from ^99^Mo/^99m^Tc Tekcis® generator. The solution was incubated at room temperature. Samples were taken at regular time intervals of 1 min, up to 20 min, and the amount of ^99m^Tc bound to the HSA was measured, according to the QC procedures.

#### 2.2.3. [^99m^Tc]Tc-NC Labelling Kinetics

According to the manufacturers’ instructions, the NC dissolved in the kit was reconstituted with 2000 MBq of sodium pertechnetate [^99m^TcO_4_^−^] obtained from ^99^Mo/^99m^Tc Tekcis® generator. The solution was incubated at room temperature. Samples were taken at regular time intervals of 1 min, up to 20 min, and the amount of ^99m^Tc bound to the NC was measured, according to the QC procedures.

#### 2.2.4. [^99m^Tc]Tc-NC Binding Affinity Studies

Samples containing 50 µg of NC dissolved in 1 mL of deionised water were incubated with increasing concentration of ^99m^Tc to reach values of about 35, 50, 100, 200, 300 nmol ^99m^Tc/mg NC. The incubation time was fixed at 20 min and the reaction was carried out at room temperature. The amount of bound ^99m^Tc was performed as described in the QC procedures. The uptake measurements were expressed in nmol ^99m^Tc bound/mg NC.

#### 2.2.5. [^68^Ga]Ga-NC Binding Affinity Studies

The procedure described above was replicated to evaluate the binding affinity of NC with [^68^Ga]Ga. At the fixed incubation times of 20 min, NC samples were incubated with increasing concentration of ^68^Ga to reach values of about 3, 6, 12, 20, 40, 60, 120 nmol ^68^Ga /mg NC. The reaction was carried out at 40 °C. The uptake measurements were expressed in nmol ^68^Ga bound/mg NC.

#### 2.2.6. Competition Evaluation between ^68^Ga and ^99m^Tc for NC Labelling

To verify the ^99m^Tc and ^68^Ga competition for the nanocolloid binding sites, a commercial kit of NC was dissolved in 10 mL of saline solution. 18 nmol of ^99m^Tc and 13 nmol of ^68^Ga were added to 1 mL of this solution and incubated for 20 min at room temperature.

The measurement of the ^99m^Tc vs ^68^Ga bound was performed after 1 h and 3 h, as described in the QC procedures.

#### 2.2.7. Effects of pH on [^99m^Tc]Tc-NC Labelling

Two methods were used to evaluate the effect of pH on the [^99m^Tc]Tc-NC radiolabelling. A commercial kit of NC was first dissolved in 10 mL of saline solution, then 6 vials were prepared with 100 µL of this suspension and, finally, 300 KBq of ^99m^Tc were added in each vial.

Method 1: the solution was incubated at pH 6 for 20 min at room temperature. At the end of incubation, the pH of 2 vials was adjusted to pH 3 and pH 10, with HCl 5 M and NaOH 2 M, respectively.

Method 2: The pH of 3 vials was immediately adjusted to reach a final pH value of 3, 6 or 10 with HCl 5 M and NaOH 2 M, and then incubated for 20 min at room temperature.

The measurement of ^99m^Tc finally bound was performed as described in the QC procedures.

#### 2.2.8. [^99m^Tc]Tc -HSA Stability Test via DMSA Competition

These experiments were carried out to test the ^99m^Tc binding strength. DMSA was chosen as a highly competitive ligand, due to its ability of heavy metal sequestration. DMSA is therefore commonly used in case of heavy metal poisoning. A high excess of DMSA was used.

Two methods were used to evaluate the stability of [^99m^Tc]Tc-HSA through competition with DMSA.

Method 1: commercial kits of DMSA and HSA previously dissolved in 4 mL of deionised water were mixed. 200 MBq of ^99m^Tc were added, and the resulting solution was incubated at pH 6 for 20 min at room temperature.

Method 2: a commercial kit of HSA was dissolved in 4 mL of deionised water and 200 MBq of ^99m^Tc were added for incubation at pH 6, for 20 min at room temperature. Then a solution of the commercial kit of DMSA dissolved in 4 mL of deionised water was added.

The measurement of ^99m^Tc finally bound to the HSA and to DMSA was performed following the above-described QC procedures.

#### 2.2.9. [^99m^Tc]Tc-NC Stability Test via DMSA Competition

Two methods were used in order to evaluate the amount of ^99m^Tc strongly bound to the NC, via competition with DMSA. A high excess of DMSA was used.

Method 1: commercial kits of DMSA and NC previously dissolved in 4 mL deionised water were mixed. 200 MBq of ^99m^Tc were added, and the resulting solution was incubated at pH 6 for 20 min at room temperature.

Method 2: a commercial kit of NC was dissolved in 4 mL of deionised water and 200 MBq of ^99m^Tc were added for incubation at pH 6, for 20 min at room temperature. Then a solution of the commercial kit of DMSA dissolved in 4 mL of deionised water was added.

The measurement of ^99m^Tc finally bound to the NC and to DMSA was performed following the above-described QC procedures.

## 3. Results

### 3.1. Quality Controls and Labelling Yield

As already stated, in this article Nanoalbumon^®^, Vasculocis^®^ and Renocis^®^ were generically referred as NC, HSA and DMSA, respectively.

### 3.2. Kinetics Study of HSA and NC Labelling with ^99m^Tc

This test was performed to investigate the kinetic order of HSA or NC labelling with ^99m^Tc, verifying any possible difference.

The labelling kinetic order of both HSA and NC with ^99m^Tc was analysed by plotting 1/ (%^99m^ Tc) vs time. In both cases, the graphs’ trend showed a linear slope, in accordance with a pseudo-second order kinetic (Figure 1). It was however important to notice that the [^99m^Tc]Tc-NC labelling reaction speed was significantly higher than that of [^99m^Tc]Tc-HSA.

### 3.3. [^99m^Tc]Tc/[^68^Ga]Ga-NC Binding Affinities

Persico et al. calculated the NC size and particle number statistically present in a vial of NanoAlbumon^®^ and the number of Human Serum Albumin molecules forming a single nanocolloid particle [12]. The further step was to evaluate the amount of ^99m^Tc and ^68^Ga binding each nanocolloid particle, and hence also each Human Serum Albumin molecule within a single nanoparticle.

These findings were carried out through Bmax and Kd values (Table 1), derived from saturation binding experiments, that also made it possible to verify the presence of an allosteric effect, according to the sigmoid shape of the curves (Figure 2).

Kd represents the ligand concentration that binds to half the receptor sites at equilibrium. From the Kd value, the number of atoms bound at equilibrium to each single NC molecule was calculated. The maximum specific binding provided from Bmax made it possible to determine the number of atoms specifically bound to each NC particle.

As shown in Table 2, at equilibrium, there was no significant discrepancy between the number of ^99m^Tc and ^68^Ga atoms bound to NC (18.38 atoms of ^99m^Tc vs 17.81 atoms of ^68^Ga). However, the number of [^99m^Tc]Tc atoms specifically bound was less than one third of ^68^Ga (6.92 atoms of ^99m^Tc vs 18.45 atoms of ^68^Ga).

### 3.4. ^68^Ga and ^99m^Tc Competition for NC

The purpose of this test was to study the ^99m^Tc and ^68^Ga competition for the same binding site of NC.

As shown in Table 3, at first the amount of ^99m^Tc bound to NC is considerably greater than ^68^Ga, but after an incubation of 3h, the molar fraction of bound/unbound atoms was the same for the two isotopes.

### 3.5. Effects of pH on [^99m^Tc]Tc-NC Labelling

The aim of these experiments was to evaluate the effect of pH on the radiolabelling yield, expressed as bound/unbound molar fraction. A better understanding of pH influence could be helpful for developing an efficient labelling procedure. In fact, amino acid residues are strongly affected by pH changes, and the de/protonation of the residues involved in isotopes binding could negatively impact on the labelling reactions [12].

The pH effect has been tested by changing the pH either after radiolabelling (Method 1), or simultaneously (Method 2). When changes occurred after the standard labelling procedure (Method 1), the molar fraction of ^99m^Tc bound to NC is not affected. The results obtained with Method 2 stated at 6 the optimal pH for labelling. At pH 3 the labelling was lower, while at pH 10 the conjugation did not occur (Table 4).

### 3.6. HSA and NC Competition with DMSA for ^99m^Tc Stability Bond

These tests were carried out to verify the strength of the bond between ^99m^Tc and HSA or NC and were executed in presence of DMSA, a strong ligand able to compete for ^99m^Tc sequestration.

DMSA was added during the [^99m^Tc]Tc-HSA/NC labelling procedure (Method 1) or immediately after (Method 2) and the ^99m^Tc rate bound to HSA/NC or DMSA was calculated after a fixed time of 20 min.

Regarding HSA, the [^99m^Tc]Tc-DMSA complex formed with Method 1 is more stable than the [^99m^Tc]Tc -HSA one (0.087 vs 0.013, respectively). In Method 2, DMSA was able to transchelate about 30% of the ^99m^Tc previously bound to HSA (0.071 vs 0.029), as shown in Table 5.

Regarding NC, the [^99m^Tc]Tc-NC complex formed with Method 1 is more stable than the [^99m^Tc]Tc-HSA one (0.096 vs 0.013, respectively). Moreover, in Method 2, DMSA was able to transchelate less than 10% of the ^99m^Tc previously bound to NC (0.091 vs 0.009), as shown in Table 6.

Further analysis reported the number of ^99m^Tc atoms not affected by the transchelation process fielded by DMSA addition of Method 2 (Table 7). Only 1.95 1 × 10^−4^
^99m^Tc atoms are tightly bound to HSA (about 1 atom/5000), while NC better withstands transchelation, with the ^99m^Tc atoms tightly bound rising to 46.4. This value was then normalised to the number of Human Serum Albumin molecules forming a single NC particle showing an increase of ^99m^Tc atoms tightly bound per HSA molecules forming a single NC particle of 4 log (2.07) (Table 7).

## 4. Discussion

### 4.1. Radiolabelling Kinetics of HSA and NC with ^99m^Tc

The order of the labelling reactions between ^99m^Tc and HSA and between ^99m^Tc and NC were analysed plotting 1/ (%^99m^Tc) vs time (Figure 1) [13]. The % activity was proportional to the concentration of atoms, thus making it possible to determine a second order kinetics for both HSA and NC, but not the rate constant. It was also interesting to note that the NC labelling speed is double compared to that of the HSA. Evaluation of this data confirms that, contrary to what is commonly stated, not all radiolabelling reactions obey a pseudo-second order kinetics, confirming the hypothesis from J.P. Holland [14,15].

### 4.2. [^99m^Tc]Tc-NC and [^68^Ga]Ga-NC Binding Affinity Studies

The amount of bound ^99m^Tc was plotted against total ^99m^Tc/mg of NC and the data were fitted by non-linear fitting, as the one site specific binding with Hill Slope equation (Equation (1)) in GraphPad Prism 5 (GraphPad Software, San Diego, CA, USA)
(1)$$Y=\mathrm{Bmax}\;\frac{X^{h}}{{\mathrm{Kd}}^{h}+X^{h}},$$
where [Y] was the concentration of bound radioligand, Bmax was the maximum of specific binding site, [X] was the total concentration of radioligand, Kd was the ligand concentration that binds to half the receptor sites at equilibrium and h was the Hill slope.

If h equals 1.0, then binding with no cooperativity to one site occurred; when h is greater than 1.0, then multiple binding sites with positive cooperativity is implied. The Hill slope is less than 1.0 when there are multiple binding sites with different affinities for ligand or when there is negative cooperativity.

NC can bind to both ^99m^Tc and ^68^Ga and both ^99m^ Tc and ^68^Ga produced a sigmoidal binding curve with NC (Figure 2), more pronounced in the interaction with ^99m^Tc. R^2^ values obtained were also coherent (Table 1). Nonlinear regression analysis showed that the Hill slope (h) is 2.158 for ^99m^Tc, thus suggesting a high positive cooperativity (positive allosteric effect) between ^99m^ Tc and NC. Positive cooperativity (h >> 1) also implying that the first radioisotope binds NC with a lower affinity than the subsequent atoms.

The Hill slope coefficient for ^68^Ga is 1.395, a value h > 1 attesting that ^68^Ga can bind NC.

The results obtained can be directly related to patients, because a proportion was made between the amount of NC, ^68^Ga and ^99m^Tc used for experiments and the activity routinely administered.

The results showed that there is no substantial difference between the number of atoms of ^99m^Tc and ^68^Ga bound to equilibrium. The amount of ^68^Ga bound to NC is twice as much when compared to ^99m^Tc, probably due to the small size and high charge of Gallium (which favours interaction with amino acidic residues of NC). Moreover, the ^99m^Tc binding needs a more specific coordination environment to occur, due to the larger dimension, low charge, octahedral structure and multiple and different oxidation state [16,17].

Regarding the larger dimensional fraction of NC distribution, of 15-30 nm, it is interesting to compare the number of ^68^Ga atoms bound to NC with the number of Human Serum Albumin forming the single NC particle [10]. It is important to note that about 22.3 albumin molecules are involved in this NC fraction, while the number of ^68^Ga atoms strongly bounded is about 18.45 (Table 2). This leads to the conclusion that ^68^Ga strongly binds to the cysteine 34 of albumin, because this residue is repeated only once in human serum albumin molecules [16]. The free thiol fragments of cysteine, present in NC as cys-34, spontaneously form disulphide bonds at basic pH [17]. This supports our previous hypothesis. In fact, ^68^Ga does not bind to pH above 7.

### 4.3. ^68^Ga and ^99m^Tc Competition for NC

This experiment was carried out to verify whether the same binding sites of NC were involved in Technetium and Gallium binding, and therefore to examine the binding competition. The comparison was made between the Technetium concentration bound to NC in presence of Gallium, using the concentration that is usually used in Nuclear Medicine (recognised to give a binding of at least 95%). The results, reported in Table 3, show that NCs initially bond more ^99m^Tc compared to ^68^Ga, but, after 3h, the bound quantity remains substantially the same for both isotopes, since it can be explained by the inertia of complexation by the colloids and by the absence of competition for the same binding sites. Using different sites, the two radioisotopes do not compete with each other.

### 4.4. Effects of pH on ^99m^Tc Radiolabelling

Protonation and deprotonation modulate the charge state of basic and acidic amino acid side chains (Lys, Arg, Glu, Asp, His, for which Tc has great affinity [18]) as well as of the protein termini (carboxyl and amino terminus) [19].

For both methods, the best pH for radiolabelling is 6, as actually reported in the radiopharmaceutical specifications.

With Method 1, the pH modification led to a decrease in the bound fraction, but not substantially, as it does with Method 2, because ^99m^Tc binding induces a closed conformation of the binding site thereby making close contact of cysteine fragments favourable for the formation of a disulphide bond. Once the disulphide bond forms, the closed conformation is more stable than the open apo conformation, leading to the slow release of the captured metal ion [20]. Therefore, the change of pH only slightly affects the binding sites, as they are sheltered.

With Method 2, the fraction varies more at pH 10, probably due to the accumulation of negative charges inside the binding pocket, preventing its binding function [21]. Naturally, this does not happen at pH 3.

### 4.5. HSA and NC Competition with DMSA for ^99m^Tc Stability Bond

The data of the experiment with Methods 1 and 2 (Table 5) for HSA, reveal that the fraction of bond ^99m^Tc is lower than that of the NC (Table 6). This behaviour is probably due to the fact that the amino acid residues are not already optimally pre-organised for binding the Technetium.

For NCs, however the behaviour changes: both in Method 1 and Method 2, ^99m^Tc binds quickly and stably to NC and there is no significant difference between Method 1 and Method 2.

For Method 2, an attempt was made to normalise this value to the number of Human Serum Albumin molecules forming a single NC particle. The quantity of HSA was calculated referring to the values of dimension of NC NanoAlbumon® obtained by Persico et al. [10] and assimilating the shape of sphere both for HSA, with a radius of 8 nm, and NC.

Table 7 shows that in a molecule of HSA into an NC particle, two ^99m^Tc atoms are tightly bound, a value sharply different from the value for free HSA, that is, 1 out of 5000. Loose bonds locate the metal too close to the solvent [20,21], facilitating its transchelation. It seems obvious that the NC shows a positive allosteric effect, otherwise such a high value would not be possible, confirming what was hypothesized in Section 4.2. *[^99m^Tc]Tc-NC and [^68^Ga]Ga-NC Binding Affinity Studies*. The authors should discuss the results and how they can be interpreted from the perspective of previous studies and of the working hypotheses. The findings and their implications should be discussed in the broadest context possible. Future research directions may also be highlighted.

## 5. Conclusions

At the dawn of Nuclear Medicine, NC tracers were developed based on the fundamental role played by albumin proteins. Albumin is an endogenous functional carrier, driving a huge amount of different molecules from human blood to the target district or organ. For this reason, the peculiar function of HSA has often been exploited for pharmacological purposes, to easily carry drugs, including radiotracers. Though the drugs based on HSA are among the most useful probes for diagnosis, the so-called shine-through phenomenon limits preoperative detection with planar scintigraphy ^99m^Tc based SPECT-CT during operative exploration. The use of PET-CT, with better spatial resolution, might circumvent this problem. In addition, the opportunity to quantify the absolute target uptake in terms of SUV (Standardised Uptake Value), allows clinicians to predict intraoperative detectability. Furthermore, the physical characteristics of the ^68^Ga isotope, given its high percentage of positron emission (89%), relatively short half-life (t_½_ = 67.71 min) and chemical properties, make it an excellent positron emission isotope with superior resolution, speed and quantification capacity, compared to ^99m^Tc labelling, particularly in guided robot-assisted surgery and robotic-arm PET/CT assisted biopsy.

Since the chemical nature of the ^99m^Tc and ^68^Ga bond with NC is not fully known, our experience confirmed that pH is an essential parameter influencing Technetium and Gallium binding to NC, with an optimal range between 4.0–6.5. In all likelihood, the protein structure of NC does not favour the binding of the metals in basic conditions. In fact, some chelator groups, such as the SH group, may not be stable under a basic condition (pH 10) and may be easily oxidised, losing their radiolabelling property. Neutralisation of the ^68^Ge/^68^Ga generator eluate might gradually increase the affinity of the various Gallium chemical species for NC. Based on our results, we assume that ^68^Ga strongly binds to cys 34, the only amino acids residue once present in HSA molecules. Due to the slight competition for the same binding sites, dual labelling (^99m^Tc and ^68^Ga) is possible without problem. The experiments were generally consistent with each other, and the ^99m^Tc -NC show a fair amount of resistance to transchelation, where NC probably incorporate ^99m^Tc in structural pockets. Thus, radiolabelling is possible with excellent efficiency, both for ^99m^Tc and ^68^Ga, through pseudo-second order kinetics. Knowing well the labelling mechanism of an HSA based drug is essential for making predictions on its behaviour with different radioisotopes. A vast increase of their use is expected in the coming years, thanks to their unique characteristics of in vivo stability, bioavailability, non-toxicity, EPR, and so forth. Further studies will be conducted, focusing on the radiolabelling at physiological pH (7.4) and under the hypoxic condition, that may modify the amino acids’ bonding properties.

## Figures and Tables

**Figure 1 nanomaterials-11-01776-f001:**
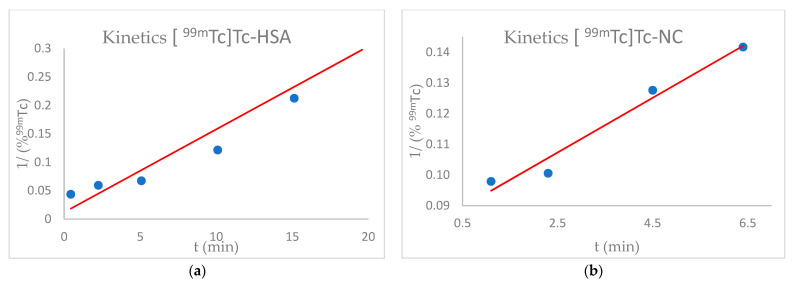
Labelling reaction between [^99m^Tc]Tc and HSA (**a**) or NC (**b**) demonstrate a pseudo-second order reaction.

**Figure 2 nanomaterials-11-01776-f002:**
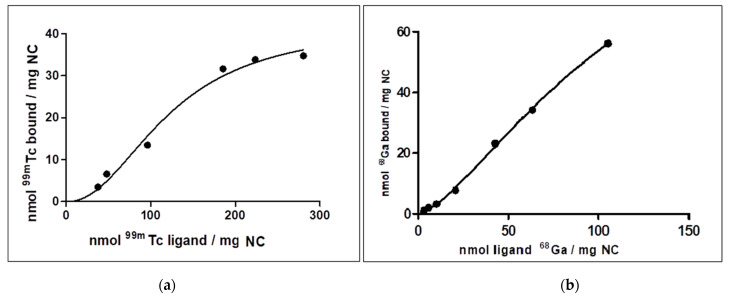
Saturation Binding experiment (one site—Specific binding determined by Hill slope equation), for binding affinity between NC and ^99m^Tc (**a**) and ^68^Ga (**b**).

**Table 1 nanomaterials-11-01776-t001:** Kd, Bmax and h-values obtained from [^99m^Tc]Tc/[^68^Ga]Ga-NC saturation binding experiments.

	Kd	Bmax	h	R2
[^99m^Tc]Tc-NC	123.1	42.32	2.158	0.9873
[^68^Ga]Ga-NC	119.4	123.7	1.395	0.9969
95% Confidence Intervals				
[^68^Ga]Ga-NC	40.30 to 198.6	58.23 to 189.2	1.158 to 1.801	
[^99m^Tc]Tc-NC	40.02 to 206.2	20.99 to 63.65	0.5075 to 3.809	

**Table 2 nanomaterials-11-01776-t002:** Number of ^99m^Tc and ^68^Ga atoms bound to a single nanocolloid particle and to the Human Serum Albumin molecules contained in each nanoparticle. Mean (SD).

NC Particles	Atoms Number	HSA Molecules Contained in a Single NC Particle	Atoms Number
^99m^Tc atoms bound at equilibrium	2.97(1) 1 × 10^10^	^99m^Tc atoms bound at equilibrium	18.4(2)
^68^Ga atoms bound at equilibrium	9.44(1) 1 × 10^4^	^68^Ga atoms bound at equilibrium	17.8(1)
^99m^Tc specifically bound	1.12(1) 1 × 10^10^	^99m^Tc specifically bound	6.9(1)
^68^Ga specifically bound	9.78(1) 1 × 10^4^	^68^Ga specifically bound	18.5(1)

**Table 3 nanomaterials-11-01776-t003:** ^99m^Tc and ^68^Ga amount bound/unbound to NC reported as molar fraction. Mean (SD).

	^99m^Tc Atoms	^68^Ga Atoms	^99m^Tc Atoms	^68^Ga Atoms
	After 1 h		After 3 h	
Bound molar fraction	7.98(1) 1 × 10^9^	1.82(1) 1 × 10^9^	1.23(2) 1 × 10^9^	1.06(1) 1 × 10^9^
Unbound molar fraction	3.19(1) 1 × 10^9^	5.98(1) 1 × 10^9^	4.02(2) 1 × 10^9^	3.91(1) 1 × 10^9^

**Table 4 nanomaterials-11-01776-t004:** Molar fraction of ^99m^Tc amount bound/unbound to NC depending on pH values. Mean (SD).

	Method 1		Method 2	
pH	bound molar fraction	unbound molar fraction	bound molar fraction	unbound molar fraction
10	0.637(1)	0.363(1)	0.005(2)	0.995(2)
6	0.912(1)	0.088(1)	0.896(1)	0.104(1)
3	0.742(2)	0.258(2)	0.632(1)	0.368(1)

**Table 5 nanomaterials-11-01776-t005:** Amount of ^99m^Tc atoms bound to HSA or to DMSA reported as molar fraction. Mean (SD).

	[^99m^Tc]Tc-HSA Molar Fraction	[^99m^Tc]Tc-DMSA Molar Fraction
Method 1	0.013(1)	0.087(1)
Method 2	0.071(2)	0.029(1)

**Table 6 nanomaterials-11-01776-t006:** Amount of ^99m^Tc atoms bound to NC or to DMSA reported as molar fraction. Mean (SD).

	[^99m^Tc]Tc-NC Molar Fraction	[^99m^Tc]Tc-DMSA Molar Fraction
Method 1	0.096(1)	0.004(1)
Method 2	0.091(1)	0.009(2)

**Table 7 nanomaterials-11-01776-t007:** ^99m^Tc tightly bound to HSA, NC and then normalised to the Human Serum Albumin that form NC. Mean (SD).

Method 2	
^99m^Tc atoms per HSA	1.95(2) 1 × 10^−4^
^99m^Tc atoms per NC	46.4(2)
^99m^Tc atoms per molecules of Human Serum Albumin in NC	2.1(2)

## Data Availability

The data presented in this studies are available on request from the corresponding author.

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
