# Peer review of "Chemical and Physical Characterisation of Human Serum Albumin Nanocolloids: Kinetics, Strength and Specificity of Bonds with 99mTc and 68Ga"

_nanomaterials, 2021, doi:10.3390/nano11071776_

Round 1
Reviewer 1 Report
Dear Authors,
In this manuscript, you showed us that in vitro result of the binding properties of radioisotopes and albumin-based nanocolloids (NC). I think you thoroughly investigated the 99mTc or 68Ga and NC binding kinetics, affinity and pH influences. All the procedures, experiments and analysis for this study were faithful and scientifically sound.
Please correct as bellows
Corrections
- In the entire manuscript, figure and figure legends, please check the nomenclature of radiolabeled particles (Nuclear Medicine and Biology 2017;55:v–xi).
99mTc-NC -> [99mTc]Tc-NC
68Ga-NC -> [68Ga]Ga-NC
Etc.
- In Introduction,
SPET-CT -> SPECT-CT
SPET -> SPECT
- In Methods,
99mTc-HAS labelling kinetics -> 99mTc-HSA labelling kinetics
Author Response
Dear Editor,
We kindly thank you for sending us the reviewers' comments to our manuscript.We totally agree with, therefore we have tried to implement our elaborate with the suggested corrections.
Hoping to have met your expectations, we well be looking for futher feedback.
Best Regards
Manuela Marenco

Reviewer 2 Report
This manuscript has focused on the radiolabeling properties of human serum albumin (HSA) and its nanocolloids (NCs) with Tc-99m and Ga-68. Different methods were used to study the radiolabeling properties including kinetics, affinity, and stability as well as related Kd, Bmax, atom-binding number, and molar fraction under different conditions. The results have revealed interesting information of pH effects, 68Ga/99mTc competitions, and competitions in the presence of a chelator DMSA, guiding further research and developments.
Questions and suggestions: 1) Effect of physiological pH: it is important to further study the radiolabeling and properties at physiological pH 7.4 in addition to pH 3, 6, and 10. 2) Effect of hypoxic condition: it is important to study the radiolabeling of Tc-99m under a hypoxic condition. Especially, some chelator groups such as SH group may not be stable under a basic condition (such as pH 10). For example, SH group may be easily oxidized under a basic condition to form disulfide bond, losing radiolabeling property. 3) Statistic values: all the data in all the tables were obtained by multiple experiments, so should be presented with statistics and related parameters (experiment times, mean values, and standard deviation)
Author Response

(The authors gave the same response as above.)
